# Adipose Tissue Dysfunction and Metabolic Diseases: The Role of Vitamin D/Vitamin D Receptor Axis

**DOI:** 10.3390/ijms262110256

**Published:** 2025-10-22

**Authors:** Flavia Agata Cimini, Federica Sentinelli, Alessandro Oldani, Ilaria Barchetta, Maria Gisella Cavallo

**Affiliations:** 1Department of Experimental Medicine, Sapienza University, 00161 Rome, Italy; flaviaagata.cimini@uniroma1.it (F.A.C.); alessandro.oldani@uniroma1.it (A.O.); gisella.cavallo@uniroma1.it (M.G.C.); 2Endocrinology and Diabetes, Department of Clinical Medicine, Public Health, Life and Environmental Sciences (MeSVA), University of L’Aquila, 67100 L’Aquila, Italy; federica.sentinelli@univaq.it

**Keywords:** vitamin D (VD), vitamin D receptor (VDR), adipose tissue dysfunction, obesity, insulin resistance, adipokines, inflammation

## Abstract

Obesity-associated adipose tissue dysfunction represents a key driver of metabolic disorders, including type 2 diabetes, cardiovascular diseases, and fatty liver disease. Emerging evidence highlights the vitamin D/vitamin D receptor (VD/VDR) axis as an important regulator of adipose tissue homeostasis. Beyond its classical role in mineral metabolism, vitamin D influences adipogenesis, inflammation, and insulin sensitivity, thereby modulating systemic metabolic health. In this review, we summarize the current understanding of the VD/VDR axis in adipose tissue biology, from molecular pathways controlling lipid turnover and immune responses to experimental and clinical evidence linking vitamin D status with obesity-related complications. We also discuss the role of genetic variability and tissue-specific VDR signaling in shaping metabolic outcomes. While results from supplementation trials remain inconsistent, maintaining adequate vitamin D levels appears crucial for the prevention of adipose tissue dysfunction and its cardiometabolic consequences. Future studies are warranted to define optimal strategies for harnessing the VD/VDR axis in therapeutic approaches to obesity and metabolic disease.

## 1. Introduction

The prevalence of metabolic diseases, including obesity, type 2 diabetes (T2D), and cardiovascular disorders, has reached alarming rates globally, significantly impacting public health systems and economies. Obesity, a major driver of these conditions, is characterized by an excess of adipose tissue, which also undergoes profound structural and functional changes in response to metabolic stress [1]. Historically considered as a passive energy reservoir, adipose tissue is now recognized as an active endocrine organ, intricately involved in regulating energy balance, inflammation, and insulin sensitivity. Dysfunctional adipose tissue contributes to the pathogenesis of obesity-related diseases through chronic low-grade inflammation and altered secretion of bioactive molecules known as adipokines [2].

Over the past decade, increasing attention has been directed toward understanding the role of vitamin D (VD) and its receptor (VDR) in adipose tissue physiology and metabolic regulation. VD deficiency has been consistently linked to obesity, insulin resistance, and cardiovascular complications, although the precise mechanisms remain under investigation [3]. Studies suggest that VD/VDR signaling influences key processes such as adipogenesis, inflammation, and mitochondrial function, positioning this axis as a potential therapeutic target for mitigating metabolic disease. This review synthesizes current evidence on the VD/VDR axis and its implications for adipose tissue function, highlighting both experimental findings and clinical perspectives.

## 2. Adipose Tissue and Metabolic Diseases

### 2.1. The Adipose Tissue as an Endocrine Organ

Adipose tissue (AT), once viewed solely as an inert organ responsible for energy storage and release, has evolved into a focal point of extensive research due to its newly uncovered multifaceted roles within the body. Traditionally thought of as passive, AT has now been recognized as a dynamic player in numerous physiological processes, functioning well beyond its historical role [2]. Over the last few decades, accumulating evidence has shown that AT acts not only as a storage site for triglycerides but also as a metabolically active endocrine organ [2]. This tissue is capable of synthesizing and releasing various adipokines, including adiponectin, leptin, resistin [4], apelin [5], adipsin, and visfatin [6], which exert wide-ranging effects on metabolism, inflammation, insulin sensitivity, and overall energy homeostasis. AT secretory functions contribute to the regulation of systemic inflammation, insulin secretion, and appetite, all of which respond to both internal metabolic signals and external environmental factors by modulating the release of pro-inflammatory and anti-inflammatory cytokines [7]. For instance, leptin regulates hunger signals and energy expenditure, while adiponectin enhances insulin sensitivity and possesses anti-inflammatory properties. Resistin, on the other hand, has been implicated in promoting insulin resistance [8]. Thus, AT not only stores excess energy but also serves as a central regulator of metabolic and inflammatory homeostasis, demonstrating its pivotal role in conditions such as obesity, insulin resistance, and type 2 diabetes [4,5,6,7].

A critical, yet often underappreciated, component of AT’s complex structure is the stromal vascular fraction (SVF), which can be regarded as the local immune milieu. The SVF comprises a variety of cell types, including mesenchymal stem cells, endothelial cells, and immune cells, such as macrophages, T cells, and eosinophils, which collectively play vital roles in maintaining tissue homeostasis [9]. These cells form an intricate network within AT, particularly around crown-like structures and perivascular areas, where they regulate immune responses and inflammation [10]. In this context, the interplay between immune cells and the sympathetic nervous system within AT is particularly intriguing. Sympathetic nerve fibers that innervate adipose tissue not only regulate lipolysis and energy expenditure but also influence immune cell activity. For example, immune cells, such as macrophages and eosinophils, can produce neurotrophic factors, modulating sympathetic tone and local nerve growth, which may further influence adipocyte function [11]. Macrophages, in particular, play a central role in determining the inflammatory state of adipose tissue. These immune cells exist in different polarization states, with M1 macrophages characterized by their pro-inflammatory phenotype and M2 macrophages known for their anti-inflammatory properties. M1 macrophages are commonly associated with metabolic dysfunction in obesity, as they secrete pro-inflammatory cytokines such as tumor necrosis factor alpha (TNFα), interleukin (IL)-1β, IL-12, and IL-23, which can disrupt insulin signaling and exacerbate systemic inflammation [12]. On the other hand, M2 macrophages, which produce anti-inflammatory cytokines like IL-10, are more prevalent in lean individuals and contribute to maintaining insulin sensitivity and adipose tissue homeostasis. Thus, the balance between M1 and M2 macrophages is crucial for determining whether adipose tissue will promote or alleviate metabolic disease [13]. In addition to macrophages, another immune cell subset that has garnered attention in recent years is the group-2 innate lymphoid cells (ILC2s) [12,13]. These cells have been found to play an essential role in supporting metabolic homeostasis within AT by promoting the expansion and activity of adipose tissue eosinophils (ATEs). ATEs, in turn, are key players in maintaining the M2 polarization of macrophages, thus fostering an anti-inflammatory environment within the tissue [14]. Moreover, the presence of white-adipose-tissue-resident multipotent stromal cells (WAT-MSCs) has been shown to further support the activity of ILC2s and ATEs, establishing a positive feedback loop that helps preserve the functional integrity of the adipose tissue [15].

The emerging understanding of the immune and endocrine functions of the AT has profound implications for metabolic health [9]. Dysregulation of this tissue’s complex network, as in obesity, can lead to chronic low-grade inflammation, commonly referred to as “meta-inflammation,” which is a hallmark of metabolic syndrome and associated diseases such as type 2 diabetes (T2D), accelerated atherosclerosis, and metabolic-dysfunction-associated steatotic liver disease (MASLD) [16,17]. The infiltration of pro-inflammatory immune cells, such as M1 macrophages, into hypertrophic AT triggers a cascade of inflammatory responses that impair insulin signaling, promote insulin resistance, and contribute to the progression of these metabolic disorders. Additionally, the altered secretion of adipokines, such as reduced adiponectin levels and increased leptin and resistin levels, further exacerbates metabolic dysregulation (Table 1) [12,18,19].

### 2.2. Sick Fat and Metabolic Impairment

The concept of “sick fat” refers to the pathological state of AT in obesity and related metabolic conditions, where its normal physiological functions are impaired [20]. Unlike healthy AT, which is primarily in charge of energy storage, insulation, and endocrine regulation, sick fat is characterized by profound dysregulation of metabolic processes. This includes excessive inflammatory signaling, fibrosis, impaired lipid storage and mobilization, and abnormal interactions between immune cells and adipocytes [21,22,23]. The chronic inflammatory environment within dysfunctional AT exacerbates insulin resistance, promotes lipotoxicity, and increases the risk for metabolic disorders, particularly T2D [21,24,25]. One of the hallmark features of sick fat is the presence of chronic low-grade inflammation, which is triggered by an imbalance in the secretion of adipokines, bioactive molecules secreted by adipose tissue. In a healthy state, adipose-derived hormones such as adiponectin and leptin regulate energy homeostasis, insulin sensitivity, and inflammation. However, in obesity, there is an increase in the secretion of pro-inflammatory cytokines, such as TNFα, IL-6, and Monocyte Chemoattractant Protein-1 (MCP-1), while the secretion of anti-inflammatory adipokines like adiponectin is reduced [21,26,27] (Figure 1).

An impaired VD/VDR axis may lead to altered immune responses, contributing to a pro-inflammatory status and metabolic dysfunction in AT.

This shift toward a pro-inflammatory state disrupts insulin signaling pathways, promoting systemic insulin resistance, a key driver of T2D [20]. At the cellular level, the immune cell profile within AT is profoundly altered in obesity. There is an infiltration of pro-inflammatory macrophages, particularly M1 macrophages, which secrete high levels of pro-inflammatory cytokines such as TNFα and IL-1β [12,18,25]. These cytokines activate signaling pathways in adipocytes that impair insulin receptor signaling, contributing to insulin resistance [24]. Conversely, the numbers of anti-inflammatory immune cells, such as regulatory T cells (Tregs) and M2 macrophages, which maintain tissue homeostasis and suppress inflammation, are significantly reduced in sick fat. The loss of these regulatory mechanisms amplifies the inflammatory response and perpetuates a chronic state of inflammation within the tissue [30]. The physical properties of adipose tissue also change in obesity, contributing to its dysfunction (Table 2) [31].

One of the most prominent changes is adipocyte hypertrophy, where individual fat cells increase in size as they store excess lipids [32]. Hypertrophic adipocytes are less efficient in storing lipids, leading to lipid spillover into non-adipose tissues, a condition known as lipotoxicity. This ectopic lipid deposition in organs such as the liver, pancreas, and skeletal muscle further contributes to insulin resistance and metabolic dysfunction [33]. Moreover, hypertrophic adipocytes experience hypoxia, or reduced oxygen availability, because the vascular system cannot adequately expand to support the growing tissue [34]. Hypoxia triggers the activation of stress pathways, leading to adipocyte apoptosis (cell death), fibrosis, and further inflammatory cell recruitment [35].

Fibrosis, or the excessive deposition of extracellular matrix (ECM) proteins, is another pathological feature of the sick fat [22]. As AT becomes fibrotic, it loses its flexibility and ability to properly expand or contract in response to energy demands. Fibrosis also creates a hostile environment for healthy adipocyte function, leading to further metabolic impairments. In both humans and rodent models of obesity, increased levels of ECM proteins such as collagen are observed in AT, and this fibrotic remodeling is closely associated with insulin resistance [36]. Fibrosis not only impairs adipose tissue expansion but also exacerbates local hypoxia and inflammation, creating a vicious cycle of AT dysfunction [36]. The interplay between local and systemic inflammation in sick fat is crucial for understanding its role in metabolic diseases. As the AT becomes inflamed, the pro-inflammatory cytokines and chemokines released into the circulation contribute to a state of systemic, subclinical inflammation [26,27]. This low-grade systemic inflammation is a common feature of metabolic syndrome and has been linked to the development of obesity-related complications such as cardiovascular disease, fatty liver disease, and T2D [37]. The ongoing inflammatory process also impairs the ability of insulin to regulate glucose uptake in peripheral tissues, further exacerbating metabolic dysfunction.

Furthermore, adipose tissue dysfunction in obesity is not uniform across all fat depots [38]. Visceral adipose tissue (VAT), which surrounds internal organs, is particularly prone to inflammation and metabolic dysregulation compared to subcutaneous adipose tissue (SAT), which is located beneath the skin [38]. VAT is more metabolically active and secretes higher levels of pro-inflammatory cytokines, making it a key contributor to systemic insulin resistance and cardiovascular risk [39,40]. Studies have shown that individuals with higher VAT accumulation are at greater risk for metabolic diseases, regardless of overall body fat percentage, underscoring the unique role of visceral fat in the pathology of sick fat [41].

In summary, sick fat represents a pathological transformation of adipose tissue driven by obesity and characterized by chronic inflammation, immune cell dysfunction, fibrosis, and impaired lipid handling. These changes not only contribute to local adipose tissue dysfunction but also have far-reaching effects on systemic metabolic health, promoting insulin resistance, T2D, and other obesity-related diseases [22,36,37]. Addressing the underlying mechanisms of adipose tissue inflammation and dysfunction may hold the key to developing more effective therapeutic strategies for managing obesity and its associated metabolic complications.

## 3. The Vitamin D/Vitamin D Receptor Axis in Metabolic Regulation

### 3.1. Vitamin D and VDR: General Overview

Vitamin D, primarily obtained through sunlight exposure or dietary intake, plays a pivotal role in calcium and phosphate metabolism. Beyond its classical role, vitamin D exerts wide-ranging effects on various tissues, including AT, immune cells, and pancreatic β-cells. The biological effects of vitamin D are mediated through the vitamin D receptor (VDR), a nuclear receptor that regulates gene transcription. The activation of VDR by its ligand, 1,25(OH)_2_D, modulates numerous physiological processes, including those involved in inflammation and glucose metabolism. Inside the cells, the biological effects of the 1,25(OH)_2_D hormone are mediated by the vitamin D receptor (VDR) that binds the vitamin D effectively at sub-nanomolar concentrations [42]. The cloning of the VDR in 1988 [43] has been a fundamental discovery in understanding the metabolic role played by vitamin D. The additional finding that clarified the broad spectrum of activities carried out by vitamin D within the whole organism came from studies in which it was observed that the VDR expression was nearly ubiquitous [44,45,46]. However, although the VDR gene expression was determined in approximately 250 human tissues and cell types [47] and more than 3% of the human genome is under direct or indirect VDR control [48], the concentration of the protein varies significantly with the highest expression in metabolic tissues, such as adipose tissue, bone, kidneys and intestine to low or absent VDR expression in erythrocytes, striated muscle cells, and Purkinje cells of the cerebellum [49]. The presence of VDR in such a large number of tissues makes it necessary to understand the mechanisms of gene regulation of VDR itself, besides comprehending how VDR regulates other genes across the genome. The regulation of VDR is influenced by environmental factors and genetic mechanisms. Previous studies by Zella et al. identified several highly conserved VDRE regions throughout the VDR gene that mediate the actions of vitamin D. By chromatin immunoprecipitation (ChIP) and ChIP DNA microarray (ChIP-chip) analyses, the authors identified these regulatory regions located in two large introns significantly distant from the gene’s transcriptional start site and an additional region located 6 kb upstream of the VDR transcription start site [50,51]. From these results, it clearly emerges that the VDR autoregulates its own expression.

Environmental factors that influence levels of circulating vitamin D such as dietary selection (oily fish, egg yolks, mushrooms and fortified milk) [52], skin exposure to UVB irradiation [53] and the use of sunscreens that absorb UVB radiation [54], obesity [55], air pollution, aging, and so on [56] also regulate VDR gene expression. The VDR gene expression may also be modulated by genetic variations.

The VDR gene is under the control of four promoters, some of which are tissue-specific, favouring the broad spectrum of functions of VDR [48]. Among the variants identified so far, the rs2228570 C > T variant, also referred to as FokI, has been demonstrated to be functional as it is located at the start sites of translation of the VDR gene, altering the ATG start codon. When the C-allele is present, an alternative start codon located at the fourth position is used, producing a VDR protein that is truncated by three amino acids. Functional studies observed that the shorter version of the protein (424 aa) has a higher transactivational capacity than the long form (427 aa) [57]. Previously, we tested the hypothesis that the rs2228570 polymorphism affecting VDR activity might be associated with type 2 diabetes and the vitamin D system. So, we genotyped the rs2228570 variant in a large cohort of Caucasian subjects with T2D and in nondiabetic controls. However, our study did not provide evidence for the association of this polymorphism with either T2D or with circulating vitamin D levels [58]. Another important polymorphism involved in VDR regulation is the rs11568820 variant (G to A nucleotide substitution) located in the promoter region of the VDR gene. The A-allele alters the functional binding site for the intestinal-specific transcription factor Cdx-2, favouring its interaction with the promoter and increasing the intestine-specific transcription of the VDR gene [59]. We have previously observed, in an association study between the rs11568820 polymorphism and T2D, that the AA genotype conferred a higher risk of T2D and that the rs11568820 variant was also associated with impaired insulin secretion [60]. We further observed that the AA genotype was associated with 2 h high-normal glucose, a marker of cardiometabolic risk, in a cohort of overweight/obese children, highlighting that the rs11568820 polymorphism predisposes individuals towards metabolic alterations not only in adulthood but also early in life [60]. Other polymorphisms, rs1544410 (BsmI), rs7975232 (ApaI), and rs731236 (TaqI), located at the 3′- end of the VDR gene, have been previously studied [61]. The 3′-untranslated region (3′-UTR) of genes is known to regulate the degradation, stability, translation, and localization of mRNAs [62]. For this reason, previous studies focused on the relation between the amount of the mRNAs and different VDR genotypes, although the results have been conflicting. One study on the rs1544410 (BsmI) polymorphism reported no difference in the amount of mRNA between the genotypes [63]. On the contrary, another study on the rs731236 T>C (TaqI) variant observed a reduction of 30% of VDR mRNA transcript with the minor C-allele, although the half-life of these two polymorphic transcripts was similar [64]. Also, haplotype studies have been performed with these three variants. Carling et al. reported significantly lower VDR mRNA levels with the baT haplotype compared to Bat [65]. From the data available, the functional impact of rs1544410 (BsmI), rs7975232 (ApaI), and rs731236 (TaqI) variants remains unclear, and it is possible that this region is in linkage disequilibrium with other sequences to regulate transcription, translation, or RNA processing. The study of the structural domains of VDR revealed that it belongs to a superfamily of nuclear receptors (NRs) that comprises 48 members that are characterized by a highly conserved DNA-binding domain (DBD) and a structurally conserved ligand-binding domain (LBD) [66]. The binding of VDR with vitamin D in the cytosol causes its phosphorylation and conformational changes, favouring its binding with any of the three retinoid X receptor (RXR) isoforms (RXRα, RXRβ, and RXRγ), which are the predominant dimerization partners of VDR. Thus, the vitamin D–VDR–RXR complex translocates to the nucleus and, through the DBD, binds specific binding motifs named vitamin D response element (VDRE) sited in the promoter region of vitamin D-dependent genes [67]. In general, the VDREs consist of 2 hexameric nucleotide half-sites separated by three nucleotides (DR3) (Haussler). The expression of vitamin D target genes is further modulated by regulating proteins referred to as co-activators (CoAs) [68] and co-repressors (CoRs) [69] that interact with the LBD of VDR protein. The CoA proteins induce the transcription of the vitamin D target genes by the remodelling of chromatin and favouring the assembly of the basal transcriptional machinery on the transcription start site of the genes [68]. On the contrary, the recruitment of co-repressor proteins [70,71] keeps chromatin in a condensed configuration that is inaccessible to the transcription protein machinery, causing gene downregulation. An additional mechanism of gene regulation controlled by VDR is the presence in some vitamin D target genes of negative vitamin D response elements (nVDREs), involved in a mechanism of transcriptional gene repression [72,73]. An overview of VDR gene structure and function is summarized in Table 3.

Previous studies have reported that the VDR gene is expressed in 3T3-L1 murine adipocytes [74] and in human pre-adipocytes and differentiated adipocytes [75,76]. Also, VDR is reported to be expressed in human subcutaneous and visceral adipose tissue [75] and in human mammary adipocytes [77]. These findings support the importance of the vitamin D/VDR axis for the metabolic processes in the adipose tissue.

### 3.2. VD/VDR in Metabolic Diseases: Experimental Evidence

There is growing experimental evidence supporting the role of the VD/VDR axis in metabolic regulation. In vitro studies have shown that VDR activation in adipocytes regulates lipid storage and inhibits pro-inflammatory cytokine production [78]. In mouse models, VDR knockout (VDRKO) mice display reduced adipose tissue mass, smaller adipocytes, and lower leptin levels, whereas treatment with 1,25(OH)_2_D induces leptin expression and secretion [79,80]. Similarly, 1,25(OH)_2_D supplementation has been shown to stimulate adiponectin secretion in 3T3-L1 murine adipocytes [78]. However, other in vitro studies on human subcutaneous adipocytes have reported an inhibitory effect of 1,25(OH)_2_D on adiponectin production [81,82], underscoring cell- and species-specific differences. In an in vivo study, performed in a diet-induced obesity mouse model, the authors observed an increase in adiponectin plasma concentration in mice fed a high vitamin D diet compared to mice subjected to a high-fat diet [83].

In primary culture of human adipocytes, vitamin D treatment suppressed mRNA levels and secretion of leptin and IL-6, suggesting the inhibition of the inflammatory pathway [84].

In human studies, clinical evidence also supports a link between the VD/VDR axis and metabolic diseases, although findings remain inconsistent across populations. Observational studies have associated low serum vitamin D levels with an increased risk of metabolic disorders [79,85], and vitamin D supplementation has been reported to influence adipokine profiles. For instance, in the NHS and HPFS cohorts, vitamin D levels were positively associated with adiponectin [86], while the META-Health Study reported a gender-, race-, and BMI-dependent relationship [87]. In obese children with vitamin D deficiency, calcitriol treatment increased adiponectin levels [88], and similar improvements were observed in adults with type 2 diabetes following vitamin D-fortified food supplementation [89]. Nonetheless, meta-analyses of randomized controlled trials did not confirm significant changes in circulating adiponectin after vitamin D supplementation [90].

### 3.3. Local Regulation of Vitamin D Metabolism Within Adipose Tissue

Among the plethora of genes regulated by the vitamin D/VDR axis, particular attention warrants the observation from previous studies that vitamin D itself modulates genes codifying for enzymes involved in the metabolism of vitamin D, such as CYP27B1 (1-hydroxylases) and CYP24A1 (25(OH)D-24-hydroxylase). The expression and activity of CYP27B1, the enzyme involved in the hydroxylation of 25(OH)D to 1,25(OH)_2_D, is tightly regulated by vitamin D [91]. In addition, the CYP24A1 enzyme, which controls the degradation of 25(OH)-D and 1,25(OH)_2_D to calcitroic acid and other inactive metabolites, is stimulated by vitamin D. In this way, the vitamin D promotes a negative feedback mechanism, thus regulating its own production essential to maintain tissue homeostasis. The CYP27B1 enzyme was found to be expressed not only in the kidney, but has been reported in rat adipose tissue and in cultured 3T3-L1 preadipocytes [85] and in human visceral and subcutaneous adipose tissue [78,83]. Also, the CYP24A1 enzyme was identified in both human visceral and subcutaneous adipose tissue [83]. The CYP2R1 gene that codifies the enzyme, necessary for the first hydroxylation in the C25 position of vitamin D, was found to be expressed in subcutaneous and visceral adipose tissue [83] as well.

All this evidence highlights that a regulation of vitamin D metabolism in adipose tissue is present and that this local production of 25(OH)D and 1,25(OH)_2_D may act for autocrine/paracrine purposes.

## 4. VD/VDR Axis and the Adipose Tissue

### 4.1. Pathways Involved in Adipose Tissue Homeostasis

The VD/VDR axis regulates crucial processes in adipose tissue homeostasis, including energy expenditure, lipid metabolism, and inflammation, with evidence emerging from numerous studies in rodent models. Unlike humans, where obesity is consistently associated with decreased plasma 25(OH)D concentrations, findings in mice are variable. For instance, some studies report no significant changes in plasma 25(OH)D levels in obese mice fed high-fat (HF) diets, whereas others document a decrease. These discrepancies could stem from differences in the HF diet composition, methods used to quantify 25(OH)D, such as immunoassays versus mass spectrometry, and, mostly, from physiological differences between humans and animal models.

Notably, earlier studies using ELISA-based quantification reported a decline in plasma 25(OH)D levels, but subsequent investigations using mass spectrometry under similar dietary conditions found no such decrease. Beyond total 25(OH)D levels, the reduction in free 25(OH)D rates has been consistently observed in obese rodents, paralleling human findings [92].

Additionally, plasma 1,25(OH)_2_D levels show inconsistent patterns in obesity, with reports of decreased, unchanged, or even increased levels, further highlighting the complexity of VD metabolism under HF diets [93]. VDR^−/−^ mice have been instrumental in elucidating the role of VD metabolism in adipose tissue. These mice exhibit resistance to diet-induced obesity, likely due to enhanced fatty acid oxidation and upregulation of uncoupling proteins (UCP1, UCP2, and UCP3) in adipose tissue, which collectively increase energy expenditure. However, these findings are not without caveats. VDR^−/−^ mice are fed calcium-rich rescue diets to mitigate secondary hyperparathyroidism, which itself can influence energy balance. Additionally, systemic VDR ablation affects multiple tissues, complicating the attribution of observed phenotypes specifically to adipose tissue. Moreover, VDR^−/−^ mice develop alopecia, which may increase energy expenditure through reduced insulation [94]. Conversely, overexpression of human VDR in mouse adipose tissue has been shown to increase body weight and fat pad mass, accompanied by reduced energy expenditure and fatty acid oxidation [95]. A brief overview of these models is shown in Figure 2.

Targeted invalidation of VDR in adipose tissue has yielded conflicting results. For example, Cre recombinase-driven VDR deletion under the FABP4 promoter increased visceral fat pad weight in females but had no effect on overall adiposity in males. Another model using adiponectin promoter-driven Cre recombinase reported no significant effect on body weight or adiposity, though a slight increase in visceral fat was noted. These conflicting outcomes underscore the complex and context-dependent role of VDR in adipose tissue biology, where factors such as sex, genetic background, and dietary context may influence results [96].

### 4.2. From Physiology to Metabolic Impairment

The progression from physiological to dysfunctional adipose tissue in obesity is closely tied to the disruption of VD and VDR functions. Studies on VD supplementation in obese rodents have revealed limited efficacy in reversing established obesity. For instance, supplementation with VD in obese mice improved adipose tissue inflammation, hepatic steatosis, and cardiac function but failed to reduce body weight or adiposity, consistent with findings in other studies [97,98,99]. However, injections of 1,25(OH)_2_D (10,000 IU vitamin D/kg diet for 16 weeks vs. LF diet vs. HF diet containing 1000 IU vitamin D/kg diet) demonstrated improved outcomes in body weight and adiposity, suggesting that active metabolites of VD may bypass obesity-related impairments in VD metabolism [100]. Preventive strategies appear more promising, with several studies documenting reductions in body weight and adiposity under VD or 1,25(OH)_2_D supplementation. These effects are thought to involve the induction of lipid catabolism, particularly in the liver and brown adipose tissue [101], as demonstrated in both rodent [102,103] and zebrafish models [104]. Nevertheless, VD insufficiency exacerbates weight gain and adiposity in rodents, underscoring the protective role of VD in preventing metabolic impairments [97,105].

In humans, observational studies consistently demonstrate an inverse relationship between serum 25(OH)D levels and markers of obesity, such as BMI, fat mass, and waist circumference, across all age groups, including children, adults, and the elderly [106,107,108,109]. This association is supported by findings that plasma 25(OH)D and 1,25(OH)_2_D levels are lower in obese individuals compared to their normal-weight counterparts.

Mechanisms proposed to explain this include VD sequestration in expanded adipose tissue, volumetric dilution, and reduced release of 1,25(OH)_2_D due to impaired isoprenaline-mediated lipolysis in subcutaneous fat during obesity [3,110,111]. Other factors, such as reduced hepatic synthesis due to secondary hyperparathyroidism and lower CYP2J2 mRNA levels in adipose tissue of obese women, may also contribute [78].

Furthermore, the efficacy of VD supplementation in obese individuals is often limited, with meta-analyses showing reduced increases in plasma 25(OH)D levels despite higher supplementation doses [112,113,114,115]. Genetic studies suggest that VDR polymorphisms may influence fat distribution and obesity risk, although the evidence remains inconclusive, and Mendelian randomization analyses suggest that obesity itself is the primary driver of reduced 25(OH)D levels [116,117,118]. Weight loss interventions, however, consistently improve 25(OH)D concentrations, with a 10 kg weight loss associated with an increase of approximately 10% compared to baseline values [119,120].

These findings highlight the intricate relationship between VD metabolism, the VDR axis, and AT. While VD and its metabolites play critical roles in maintaining adipose tissue physiology, their disruption in obesity exacerbates metabolic impairments, necessitating further investigation into personalized VD-based therapeutic strategies for obesity and metabolic disorders [78,79,83,102,104].

## 5. Conclusions and Perspectives

Over the past decade, growing experimental and clinical evidence has revealed the multifaceted role of the vitamin D/vitamin D receptor (VD/VDR) axis in adipose tissue biology and metabolic regulation. Beyond its classical involvement in calcium and phosphate homeostasis, vitamin D modulates adipogenesis, lipid metabolism, and inflammation, thereby influencing systemic insulin sensitivity and overall metabolic health. Studies in rodent models and adipocyte cultures have demonstrated that VDR activation improves lipid handling and reduces pro-inflammatory cytokine expression, supporting its contribution to adipose tissue homeostasis.

However, despite promising preclinical findings, clinical trials investigating the effects of vitamin D supplementation on obesity and related metabolic disorders have yielded inconsistent results. These discrepancies likely reflect methodological limitations, such as heterogeneous baseline 25(OH)D levels, insufficient supplementation doses, variable treatment durations, and lack of control for nutritional and environmental confounders. Future randomized controlled trials should therefore focus on individuals with verified vitamin D deficiency, employ appropriately high doses to achieve meaningful changes in serum 25(OH)D concentrations, and ensure standardized study designs to better assess the true metabolic impact of vitamin D repletion.

While the therapeutic efficacy of vitamin D supplementation in established obesity remains uncertain, its preventive potential is supported by observational studies linking adequate vitamin D levels to improved metabolic outcomes and reduced risk of adipose tissue dysfunction. Maintaining physiological 25(OH)D concentrations may thus represent an important component of metabolic health and prevention strategies against obesity-associated complications. Further mechanistic and clinical research is warranted to clarify the contexts and molecular pathways through which modulation of the VD/VDR axis can be effectively harnessed for metabolic and adipose tissue regulation.

## Figures and Tables

**Figure 1 ijms-26-10256-f001:**
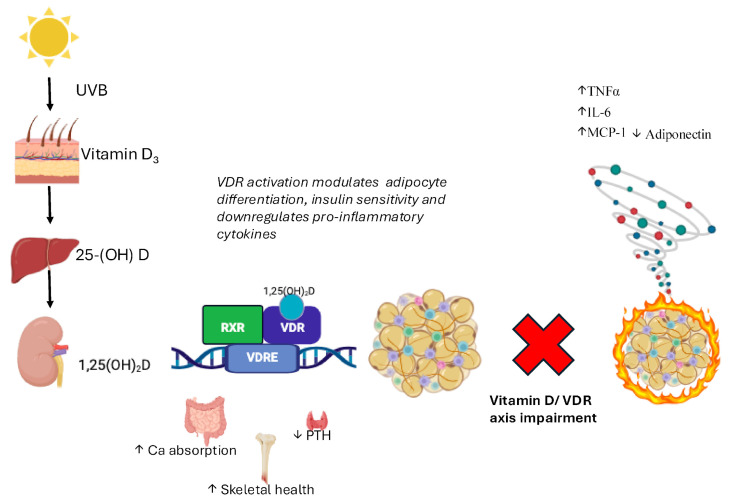
Vitamin D metabolism and VDR-mediated regulation in adipose tissue. Vitamin D_3_, synthesized in the skin upon UVB exposure, is hydroxylated in the liver to 25(OH)D and in the kidney to the active form 1,25(OH)_2_D (calcitriol). This metabolite binds to the vitamin D receptor (VDR), a nuclear receptor expressed in adipose tissue, where it modulates gene transcription [28]. VDR activation influences adipocyte differentiation, insulin sensitivity, and the inflammatory profile of adipose tissue by downregulating pro-inflammatory cytokines (TNFα, IL-6, and MCP-1) and upregulating adiponectin, thereby contributing to metabolic homeostasis [29]. **Legend:** Arrows: ↑ increase; ↓ decrease.

**Figure 2 ijms-26-10256-f002:**
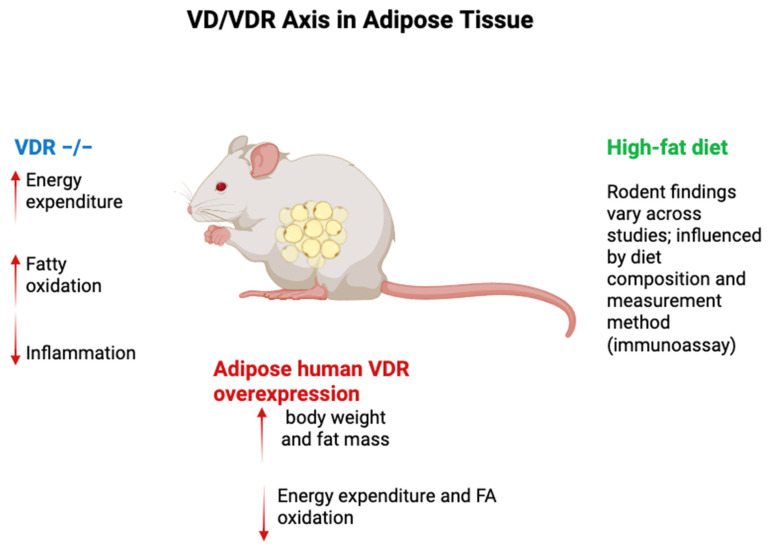
VD/VDR axis in rodent adipose tissue. In rodents, HF-diet effects on total 25(OH)D are inconsistent—driven by diet composition and assay—whereas free 25(OH)D consistently falls; 1,25(OH)_2_D responses vary. VDR^−/−^ mice show ↑ energy expenditure and FA oxidation with caveats (rescue diet, systemic deletion, alopecia). Adipocyte VDR overexpression increases fat mass/weight and lowers energy expenditure/FA oxidation; adipose VDR deletions yield promoter-/sex-/context-dependent results. **Legend:** Blue = VDR^−/−^; Red = adipocyte human VDR overexpression; Green = high-fat diet (assay & diet dependent). Arrows: ↑ increase; ↓ decrease. Abbreviations: VD, vitamin D; VDR, vitamin D receptor; FA, fatty acid.

**Table 1 ijms-26-10256-t001:** Adipose tissue functions and biological role in metabolism.

Function	Description	Key Molecules/Cells
Energy Storage and Release	Storage of triglycerides and regulated lipolysis to provide energy when needed	Triglycerides, Lipolysis pathways
Endocrine Function	Secretion of adipokines that influence systemic metabolism, appetite, and insulin sensitivity [4,5,6]	Leptin, Adiponectin, Resistin, Apelin, Visfatin, Adipsin
Metabolic Regulation	Modulation of insulin sensitivity, energy expenditure, and glucose homeostasis [4,8]	Adiponectin (↑ sensitivity), Leptin, Resistin (↓ sensitivity)
Neuro-immune Interaction	Sympathetic innervation influences lipolysis and immune activity; immune cells secrete neurotrophic factors influencing sympathetic tone [7]	Sympathetic nerves, Neurotrophic factors, Macrophages, Eosinophils
Immune Cell Niche	Stromal vascular fraction (SVF) supports mesenchymal, endothelial, and immune cells, forming a regulatory microenvironment [2]	SVF, MSCs, Endothelial cells

**Legend:** Arrows: ↑ increase; ↓ decrease.

**Table 2 ijms-26-10256-t002:** Dysfunctional adipose tissue and immune inflammation, an overview.

Function	Description	Key Molecules/Cells
Inflammatory Signaling	Release of cytokines and regulation of local and systemic inflammation Balance of M1 (pro-inflammatory) and M2 (anti-inflammatory) macrophages determines inflammatory status and insulin sensitivity [12,18,24,25,30]	IL-6, TNFα, IL-10, IL-1β, IL-12, IL-23, M1: TNFα, IL-1β, M2: IL-10
Immune modulation	Interaction with resident immune cells (macrophages, eosinophils, ILC2s, T cells) that modulate inflammation and tissue homeostasis. Maintenance of metabolic homeostasis via ILC2-induced eosinophil activation, which promotes M2 macrophage polarization [14,24]	M1/M2 macrophages, ILC2s, ATEs, T cells M1: TNFα, IL-1β M2: IL-10
Pathophysiology in obesity	Dysfunctional adipokine secretion and immune infiltration lead to chronic low-grade inflammation (“metaflammation”) and metabolic disease progression [4,5,6]	↓ Adiponectin, ↑ Leptin/Resistin, ↑ M1 macrophages

**Legend:** Arrows: ↑ increase; ↓ decrease.

**Table 3 ijms-26-10256-t003:** VDR gene: detailed overview.

Category	Details
VDR Expression	Expression nearly ubiquitous across ~250 human tissues/cell types [47]. Highest protein levels in adipose tissue, bone, kidneys, intestine [43,44]. low/absent in erythrocytes, striated muscle, Purkinje cells [49].
Genome Control	>3% of the human genome under direct or indirect VDR control [48].
Autoregulation (VDREs)	Highly conserved VDRE regions in two large introns and 6 kb upstream of TSS. Allow VDR to autoregulate its own expression [50,51].
Promoters	Four promoters control VDR transcription, some tissue-specific, contributing to functional diversity [48].
Environmental Regulators	UVB exposure increases VDR expression [53]; sunscreens decrease it [54]. Dietary vitamin D intake influences VDR levels [52]. Obesity, air pollution, and aging also modulate expression. [55,56]
Adipose Tissue Expression	VDR expressed in 3T3-L1 adipocytes, human pre-adipocytes, differentiated adipocytes, subcutaneous/visceral AT, and mammary adipocytes [74]. Highlights the role of vitamin D/VDR in adipose inflammation and metabolism [75].
Structural Domains	VDR belongs to the nuclear receptor superfamily with the conserved DNA-binding domain (DBD) and ligand-binding domain (LBD).
Polymorphism rs11568820 (Cdx2)	G > A in promoter; A-allele enhances Cdx-2 transcription factor binding, increasing intestine-specific VDR transcription [66]. AA genotype linked to higher T2DM risk and impaired insulin secretion, and early life cardiometabolic alterations [59,60].
Polymorphisms BsmI/ApaI/TaqI	Located in 3′-UTR; studies show conflicting effects on mRNA stability and transcript levels; potential linkage with other regulatory sequences [63,64,65].
Polymorphism rs2228570 (FokI)	C > T at start codon; C-allele uses downstream ATG yielding shorter VDR (424 aa) with higher transactivation compared to long form (427 aa) [57,58].
Co-regulators & nVDREs	Co-activators (CoAs) remodel chromatin and promote transcription [68]; co-repressors (CoRs) condense chromatin to repress genes [69]. Negative VDREs in some targets mediate transcriptional repression [72].

## Data Availability

No new data were created or analyzed in this study. Data sharing is not applicable to this article.

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
