# Peer review of "Adipose Tissue Dysfunction and Metabolic Diseases: The Role of Vitamin D/Vitamin D Receptor Axis"

_ijms, 2025, doi:10.3390/ijms262110256_

Round 1

Reviewer 1 Report

Comments and Suggestions for Authors

Manuscript titled “Adipose Tissue Dysfunction and Metabolic Diseases: The Role of Vitamin D/Vitamin D Receptor Axis” reports a review of various aspects of the role of vitamin D and its receptor on health, specifically, on obesity and its role in the adipose. There are some comments and suggestions for the authors:

  1. In line 79, is “dfdf” an abbreviation? After searching I could only find a DFDF domain mentioned in the literature, please clarify.
  2. Lines 114 – 130 discuss the concept of sick fat, and cite Figure 1 at the end of this paragraph. However, the title and content of this figure mention vitamin D metabolism, thus, the text and the figure do not coincide with each other, please modify as appropriate.
  3. Also in Figure 1, please eliminate the underlined text.
  4. Lines 133 – 141 mention vitamin D3, while the following paragraphs (lines 142 – 195) and Table 2 continue discussing sick fat, as mentioned in the title of this section. Are these lines correctly placed here, or should they be placed in the following section where vitamin D is the main focus? Please check the continuity of this section.
  5. The term “1,25(OH)₂D” is first mentioned in line 134, “1,25-dihydroxyvitamin D (calcitriol)” in line 203, and “1α,25(OH)2D3” in line 205. These appear to be different names of the same compound, if this is the case, is it possible to homogenize using a single term throughout the document?
  6. In line 273, please include the full name of the RXR transcription factor.
  7. Line 335 mentions the role of “vitamin D”. As currently written, it could be understood that any form of the vitamin has this role, is this correct or do you mean a specific molecule?
  8. Line 348 – 355 discuss the differences in humans and mice regarding vitamin D levels in obesity, and mention some potential reasons for the discrepancy. In addition to the factors mentioned by the authors, inherent physiological differences in human versus animal models could also be considered.
  9. Lines 394 and 398 mention supplementation and injection of vitamin D. Can you please specify the doses and timeframes of these studies?
  10. Line 426 mentions a 6 nmol/L increase, is it possible to express this value as percentage change from basal, or at least indicating the participant’s initial values? This may provide better context to the 6 nmol/L increase.
  11. The conclusion is particularly long, and does not appear congruent with the title and abstract. Thus, please consider being more succinct (some text may be moved to another section where appropriate) and restructuring such that the title, abstract, and conclusion are more in line with each other.
  12. In abbreviations, PPAR is apparently not used in the main text, please confirm and remove this abbreviation if necessary.
  13. Please remove the last two references, since these appear to be from the journal’s template.

Author Response

Reviewer 1

Manuscript titled “Adipose Tissue Dysfunction and Metabolic Diseases: The Role of Vitamin D/Vitamin D Receptor Axis” reports a review of various aspects of the role of vitamin D and its receptor on health, specifically, on obesity and its role in the adipose. There are some comments and suggestions for the authors:

In line 79, is “dfdf” an abbreviation? After searching I could only find a DFDF domain mentioned in the literature, please clarify.

"Dfdf" is a typo; we apologize for this error and have removed it from the manuscript (page 2 line 82).

Lines 114 – 130 discuss the concept of sick fat, and cite Figure 1 at the end of this paragraph. However, the title and content of this figure mention vitamin D metabolism, thus, the text and the figure do not coincide with each other, please modify as appropriate.

We thank the reviewer for this comment. Figure 1 has now been put under the specific paragraph concerning sick fat and metabolic impairment (Page 4).

Also in Figure 1, please eliminate the underlined text.

The underlined text has now been eliminated, as suggested by the reviewer.

Lines 133 – 141 mention vitamin D3, while the following paragraphs (lines 142 – 195) and Table 2 continue discussing sick fat, as mentioned in the title of this section. Are these lines correctly placed here, or should they be placed in the following section where vitamin D is the main focus? Please check the continuity of this section.

We thank the reviewer for the suggestion, and we agree that those lines belonged to the following section. We modified the manuscript accordingly. (page 6 lines 217-218)

The term “1,25(OH)₂D” is first mentioned in line 134, “1,25-dihydroxyvitamin D (calcitriol)” in line 203, and “1α,25(OH)2D3” in line 205. These appear to be different names of the same compound, if this is the case, is it possible to homogenize using a single term throughout the document?

We have now modified the manuscript and use only one name for the compound (“1,25(OH)₂D”), as suggested by the reviewer.

In line 273, please include the full name of the RXR transcription factor.

The full name of the RXR transcription factor is now included in the manuscript. (page 8, line 287)

Line 335 mentions the role of “vitamin D”. As currently written, it could be understood that any form of the vitamin has this role, is this correct or do you mean a specific molecule?

We thank the reviewer for this insightful comment. We apologize for the unclear phrasing. Our intention was to specifically refer to the active form of vitamin D, 1,25(OH)₂D (calcitriol). We have now restructured the entire chapter and provided clarification (page 9, lines 312–339; see also our response to Comment 5 from Reviewer 2).

Line 348 – 355 discuss the differences in humans and mice regarding vitamin D levels in obesity, and mention some potential reasons for the discrepancy. In addition to the factors mentioned by the authors, inherent physiological differences in human versus animal models could also be considered.

We thank the reviewer for this important observation. We have now clarified in the revised version of this manuscript that the primary reason for the observed differences in vitamin D levels between humans and mice in the context of obesity may stem from inherent physiological differences between the species. (page 11, lines 371-372)

Lines 394 and 398 mention supplementation and injection of vitamin D. Can you please specify the doses and timeframes of these studies

We thank the reviewer for this suggestion. The text has been revised to include the specific dose and treatment duration for clarity. (page 12, lines 419-420)

Line 426 mentions a 6 nmol/L increase, is it possible to express this value as percentage change from basal, or at least indicating the participant’s initial values? This may provide better context to the 6 nmol/L increase.
We thank the reviewer for the comment. The original study did not report the increase as a percentage; however, based on the mean baseline values (54.5 nmol/L) provided, we estimated the corresponding percentage change and included it in the revised version for clarity. (page 13 lines 447-448)

The conclusion is particularly long, and does not appear congruent with the title and abstract. Thus, please consider being more succinct (some text may be moved to another section where appropriate) and restructuring such that the title, abstract, and conclusion are more in line with each other.

We thank the reviewer for the helpful comment. The conclusion has been shortened and restructured to align more closely with the title and abstract, focusing on the key concepts and removing redundant details.

In abbreviations, PPAR is apparently not used in the main text, please confirm and remove this abbreviation if necessary. 

We removed the abbreviation, as suggested.

Please remove the last two references, since these appear to be from the journal’s template.

We apologize for this error, we removed the last two references.

Reviewer 2 Report

Comments and Suggestions for Authors

This review provides a well-written and substantiated overview of the role of vitamin D, its receptor and regulation as an underlying/regulating factor of adipose tissues dysfunction, obesity and metabolic disease. The manuscript addresses an overlooked topic of Vitamin D's role as an active molecule easily accessible for human physiology for regulating metabolic functions and adipose tissue. While the manuscript does present a textbook kind of structure to the topic, it is acceptable as the association of vitamin D intake and the states of fat metabolism/obesity is well substantiated with relevant clinical studies as well as in vitro and animal model studies.
However, at certain points in the manuscript these observations (clinical versus in vitro/animal) should be presented in a more separated way (see below).
In general, there is too little cited work in the introduction, as well as that the citations throughout the manuscript could be earlier positioned in the sentences/narrative (e.g., in section lines 91/112). Accumulating citations at the end of a section should be avoided (even for a review), instead citations should be used to substantiate certain parts within a sentence.
The manuscript would become more relevant if more attention were paid to the content of the Figures and Tables and its description in the main body of the text. Basically, both the graphics and tables in the manuscript are relevant but suffer from not being fully integrated in the text.
The graphics full of basic regulatory processes should be improved by handling the illustrated processed also in the main body text. Graphics should have a subscript to the illustration, basically shortly explaining the processes. The key sentences should then be described in the main body of the manuscript.
The tables should be improved by providing relevant titles, while the listings in it should have identical references as in the reference list, and the main results of each study should be understandable written in short bullet points per each study. Each of the tables should provide a column with citations. Again, the tables should be an overview/summary of the descriptions in the main text, so the main text should refer to the table, and thus table/text needs better integration.
To improve readability, especially the inclusion and positioning of citations should be improved as well that certain sections are too long or mixed in supportive evidence, and could use a cutting down in paragraphs and adding titles (examples are given below).
Lines 41 through 44 should include citations, even though most of the potential citations here will be dealt with further on in the review.
Lines 54/58 should include citations.
Lines 77/78 should include citations.
Lines 85/91 should include citations.
In line 112, the citations should be added to the descriptions of Table 1.
In line 130, the citations should be used in the previous text, and not refer to Figure 1.
Add citations to lines 133/134 through 139.
In line 153, citation #24 should be used in the text. While Table 2, should be completed by the relevant citations.
Start a new paragraph in line 179, dealing with non-uniformity dysfunction.
Add citations to lines 198/204.
Line 220 ‘Zella et al.’ instead of ‘Zella LP and co-workers’
Start a new paragraph in line 226, dealing with environmental factors influencing expression.
Start a new paragraph in line 231, dealing with promotors.
Line 288, use the citations in the previous text.
Add a number to the table named ‘VDR Gene: Detailed Overview’, to be consistent references should be numbered?
To improve readability, for section containing lines 231/289 it would be wise to make a separate header and include separate paragraphs that better refer to the Table named ‘VDR Gene: Detailed Overview’.
Section with lines 319/327 is a mixture of clinical, in vitro and animal studies, and it is strongly advised to condense the cited human studies as in the previous section with line 309/318 (of which the first lines could include more citations), and make a separate paragraph dealing with in vitro and animal observations that should include lines 303/307.
Section with lines 328/342 might be having a separate header?
Section with lines 348/391 would benefit from strictly being focused on the rodent animal model, refer to Figure 2 throughout the text, and include better citations throughout (e.g., line 350 ‘in rodent models’; line 352/353; lines 356/358; line 375; line 377 ‘another model’).
Section with lines 393/407, make sure that the studies cited are all rodent models.
Section with lines 408/413, perhaps the 4 citations could be more specified in the text, distributed and repeated over the specific results and subpopulations? 

Author Response

Reviewer 2

This review provides a well-written and substantiated overview of the role of vitamin D, its receptor and regulation as an underlying/regulating factor of adipose tissues dysfunction, obesity and metabolic disease. The manuscript addresses an overlooked topic of Vitamin D's role as an active molecule easily accessible for human physiology for regulating metabolic functions and adipose tissue. While the manuscript does present a textbook kind of structure to the topic, it is acceptable as the association of vitamin D intake and the states of fat metabolism/obesity is well substantiated with relevant clinical studies as well as in vitro and animal model studies.
However, at certain points in the manuscript these observations (clinical versus in vitro/animal) should be presented in a more separated way (see below).

In general, there is too little cited work in the introduction, as well as that the citations throughout the manuscript could be earlier positioned in the sentences/narrative (e.g., in section lines 91/112). Accumulating citations at the end of a section should be avoided (even for a review), instead citations should be used to substantiate certain parts within a sentence.

We thank the reviewer for this helpful comment. Citations have been redistributed to better support specific statements and improve the narrative flow. In some cases, grouped references were retained to preserve readability and coherence of the section. (page 3 lines 95- 104)

The manuscript would become more relevant if more attention were paid to the content of the Figures and Tables and its description in the main body of the text. Basically, both the graphics and tables in the manuscript are relevant but suffer from not being fully integrated in the text.

We thank the reviewer for the comment. The text has been revised to better integrate Figures and Tables within the main narrative. References to these elements have been added and clarified throughout the manuscript to strengthen their contextual relevance and improve readability (Figure 1 page 4, Figure 2 Page 11, Table 3, pages 8-9).

The graphics full of basic regulatory processes should be improved by handling the illustrated processes also in the main body text. Graphics should have a subscript to the illustration, basically shortly explaining the processes. The key sentences should then be described in the main body of the manuscript.

We completely agree with the suggestion and we modified the graphics accordingly.

The tables should be improved by providing relevant titles, while the listings in it should have identical references as in the reference list, and the main results of each study should be understandable written in short bullet points per each study. Each of the tables should provide a column with citations. Again, the tables should be an overview/summary of the descriptions in the main text, so the main text should refer to the table, and thus table/text needs better integration.
We thank the reviewer for this constructive suggestion. All tables have been revised accordingly. Each table now includes a clear title and we added citations identical to those in the reference list. The tables have been further integrated into the main text, which now explicitly refers to them to ensure consistency and improve readability.

To improve readability, especially the inclusion and positioning of citations should be improved as well that certain sections are too long or mixed in supportive evidence, and could use a cutting down in paragraphs and adding titles (examples are given below).
Lines 41 through 44 should include citations, even though most of the potential citations here will be dealt with further on in the review.
Lines 54/58 should include citations.
Lines 77/78 should include citations.
Lines 85/91 should include citations.
In line 112, the citations should be added to the descriptions of Table 1.
In line 130, the citations should be used in the previous text, and not refer to Figure 1.

Add citations to lines 133/134 through 139.
In line 153, citation #24 should be used in the text. While Table 2, should be completed by the relevant citations.
Start a new paragraph in line 179, dealing with non-uniformity dysfunction.
Add citations to lines 198/204.
Line 220 ‘Zella et al.’ instead of ‘Zella LP and co-workers’
Start a new paragraph in line 226, dealing with environmental factors influencing expression.
Start a new paragraph in line 231, dealing with promotors.
Line 288, use the citations in the previous text.
Add a number to the table named ‘VDR Gene: Detailed Overview’, to be consistent references should be numbered?

To improve readability, for section containing lines 231/289 it would be wise to make a separate header and include separate paragraphs that better refer to the Table named ‘VDR Gene: Detailed Overview’.
Section with lines 319/327 is a mixture of clinical, in vitro and animal studies, and it is strongly advised to condense the cited human studies as in the previous section with line 309/318 (of which the first lines could include more citations), and make a separate paragraph dealing with in vitro and animal observations that should include lines 303/307.
Section with lines 328/342 might be having a separate header?
Section with lines 348/391 would benefit from strictly being focused on the rodent animal model, refer to Figure 2 throughout the text, and include better citations throughout (e.g., line 350 ‘in rodent models’; line 352/353; lines 356/358; line 375; line 377 ‘another model’).
Section with lines 393/407, make sure that the studies cited are all rodent models.
Section with lines 408/413, perhaps the 4 citations could be more specified in the text, distributed and repeated over the specific results and subpopulations? 

The manuscript has been revised according to this reviewer’s suggestions: citations have been moved earlier in sentences to support specific statements; overly long sections were divided; and subheadings have been added to improve readability and structure.

Specific changes include:

Lines 41–44 : citations added.

Lines 54–58 : citations added.

Lines 77–78: citations added.

Lines 85–91: citations added.

Line 112: citations added in Table 1.

We modified positioning and citations relative to figure 1, as suggested.

Lines 133–139 (now page 6 lines 137-144) where meant to be a description to figure 1, we now modified the manuscript accordingly and included new citations.

Line 153: citation #24 explicitly incorporated in the text.

Table 2: completed with relevant citations.

Page 6, line 189 new paragraph introduced to discuss non-uniformity dysfunction.

Page 6 lines 209-213: citations inserted.

Line 220 (now page 7, line 231): corrected to “Zella et al.”

Page 7 line 238: new paragraph on environmental factors influencing expression.

Page 7 line 243: new paragraph on promoter regulation.

Page 8 line 293: citations moved earlier in the paragraph.

Table “VDR Gene: Detailed Overview”: now numbered as Table 3 with harmonized citations.

Lines 231–289: new subheader added (“VDR Gene: Structure, Regulation, and Variants”), with explicit references to Table 3.

We thank the reviewer for this helpful comment. Chapter 3.2 has been restructured to clearly separate the description of in vitro and animal studies from human studies, thereby improving clarity and coherence within the section.

Lines 328–342: new subheader added (“3.3 Local Regulation of Vitamin D Metabolism Within Adipose Tissue”).

Page 11 lines 364-392: we confirm that all studies discussed in this section refer to rodent models. The term “another model” has been clarified to indicate a different experimental approach within the same species, and references to Figure 2 have been added throughout the text to improve clarity and consistency.

Lines 393–407: verified that all cited studies involve rodent models.

Page 13 lines 422-427: citations redistributed and specified throughout, linked to subpopulations and outcomes.

Round 2

Reviewer 1 Report

Comments and Suggestions for Authors

Manuscript titled “Adipose Tissue Dysfunction and Metabolic Diseases: The Role of Vitamin D/Vitamin D Receptor Axis” reports a literature review regarding the various roles of vitamin D on health. The present version of the document was modified according to comments and suggestions made during an initial revision; those made by the present reviewer include:

  1. Confirming the meaning of “dfdf” in the introduction. The authors confirm that this was a typo and corrected it.
  2. Confirming the placement of Figure 1. The figure was moved to the following section.
  3. Eliminating underlined text in Figure 1. The text was corrected.
  4. Confirming the continuity of lines 133 – 141 and 142 – 195. The text was modified accordingly.
  5. Homogenizing to using a single term for vitamin D. The authors homogenized to “1,25(OH)₂D”.
  6. Including the full name of the transcription factor RXR. The full name has been added.
  7. Confirming if the authors mean vitamin D in general or a specific form of it in line 335. The authors confirm that their intention is to mean only calcitriol; the text was modified.
  8. Including inherent physiological differences between humans and animals regarding the effects of vitamin D. These differences have been specified.
  9. Specifying doses and timeframes reported in a cited paper. These values have been specified.
  10. Mentioning a numerical increase referenced as percentage for increased clarity. The authors provided an estimated percentage for this value.
  11. Restructuring the conclusion to better coincide with the title and abstract. The conclusion was modified accordingly.
  12. Removing an unused abbreviation. It was deleted.
  13. Removing some references left over from the template. They were removed.

According to the aforementioned changes made by the authors, it is apparent that they adequately considered and addressed all comments and suggestions made by the present reviewer. As a final suggestion, please check figure 1 before final publication, since it appears to be pixelated or out of focus. Thank you for your responses.

Author Response

According to the aforementioned changes made by the authors, it is apparent that they adequately considered and addressed all comments and suggestions made by the present reviewer. As a final suggestion, please check figure 1 before final publication, since it appears to be pixelated or out of focus. Thank you for your responses.

Thank you for your kind feedback. The graphical quality of Figure 1 has been ameliorated in the new version of the manuscript.

Reviewer 2 Report

Comments and Suggestions for Authors

The authors improved the general citation and flow of the paper by adding relevant explanations, separating in vitro/animalstudies from clinical data, integrating the content of the figures and tables in the narrative and adding separate headers. While the paper can be accepted a few remaining issues need to be solved first:

Table 1: remove the word ‘CITATIONS’

Table 3: choose a better wording/description for ‘Aspect’

Tables 2 and 3: place the citations correctly to each statement. Like this, Table 3 could be reduced in size by omitting the last column.

Author Response

The authors improved the general citation and flow of the paper by adding relevant explanations, separating in vitro/animal studies from clinical data, integrating the content of the figures and tables in the narrative and adding separate headers. While the paper can be accepted a few remaining issues need to be solved first:

Table 1: remove the word ‘CITATIONS’

Table 3: choose a better wording/description for ‘Aspect’

Tables 2 and 3: place the citations correctly to each statement. Like this, Table 3 could be reduced in size by omitting the last column.

Thank you for your thorough review and valuable suggestions. The word “CITATIONS” was removed from Table 1. We changed the word “aspect” with the word “Category” in Table 3. We placed citations to each statement in both Tables 2 and 3 and removed the last column of Table 3, as suggested by the reviewer.